# Robust and flexible platform for directed evolution of yeast genetic switches

Masahiro Tominaga [1], Kenta Nozaki[1], Daisuke Umeno [2], Jun Ishii [1,3 ✉] & Akihiko Kondo [1,3,4,5]

A wide repertoire of genetic switches has accelerated prokaryotic synthetic biology, while eukaryotic synthetic biology has lagged in the model organism *Saccharomyces cerevisiae*. Eukaryotic genetic switches are larger and more complex than prokaryotic ones, complicating the rational design and evolution of them. Here, we present a robust workflow for the creation and evolution of yeast genetic switches. The selector system was designed so that both ON- and OFF-state selection of genetic switches is completed solely by liquid handling, and it enabled parallel screen/selection of different motifs with different selection conditions. Because selection threshold of both ON- and OFF-state selection can be flexibly tuned, the desired selection conditions can be rapidly pinned down for individual directed evolution experiments without a prior knowledge either on the library population. The system's utility was demonstrated using 20 independent directed evolution experiments, yielding genetic switches with elevated inducer sensitivities, inverted switching behaviours, sensory functions, and improved signal-to-noise ratio (>100-fold induction). The resulting yeast genetic switches were readily integrated, in a plug-and-play manner, into an AND-gated carotenoid biosynthesis pathway.

[1] Graduate School of Science, Technology and Innovation, Kobe University, Kobe, Japan. [2] Department of Applied Chemistry and Biotechnology, Faculty of Engineering, Chiba University, Chiba, Japan. [3] Engineering Biology Research Center, Kobe University, Kobe, Japan. [4] Department of Chemical Science and Engineering, Faculty of Engineering, Kobe University, Kobe, Japan. [5] Center for Sustainable Resource Science, RIKEN, Yokohama, Japan. ✉email: junjun@port.kobe-u.ac.jp

Yeasts are attractive host organisms for synthetic biological applications, including the production of value-added chemicals[1] pharmaceutical proteins[2], sensor applications, and the reconstruction and analysis of artificial regulatory systems[3,4]. To further advance yeast synthetic biology, however, expansion of the repertoire of gene expression controllers is desirable. In particular, inducer-responsive gene expression systems (genetic switches) are eagerly awaited for precise and parallel modulation of multiple genes to program cellular behaviour as designed[5]. However, even one of the best-studied yeasts, *Saccharomyces cerevisiae*, suffers from a lack of genetic switches[6], in contrast to the model prokaryotic organism *Escherichia coli*[5].

Synthetic transcription switches in eukaryotes can be created by fusing bacterial transcription factors (bTFs) and eukaryotic transcription activators (eTAs). With the appropriate design parameters, the resulting fusion protein can act as a synthetic transcriptional activator (sTA) of a synthetic promoter (synP) containing bTF-binding sites and core promoter elements[7–11] (Fig. 1a). Most of the resultant systems perform poorly, if not nonfunctional, without finding the appropriate expression level of the sTA[12], the optimal operator arrangement[11], and the correct core promoter sequences[13]. This tuning process for artificial genetic switches is far more complicated than that needed for prokaryotic cells, because the behaviours of eukaryotic promoter systems can be easily affected by changes in surrounding sequences[14] in highly unpredictable manner. In some cases, even the ligand sensitivity of a genetic switch can be altered by several orders of magnitude[15]. Thus, to achieve a performance comparable to that of prokaryotic switches (in terms of signal-to-noise (S/N) ratio and inducer sensitivity)[5], eukaryotic systems require extensive and multi-year efforts in expression tuning and/or protein engineering of sTAs, and sequence rearrangement of the synP, as was the case for the widely used tetracycline-inducible systems (Tet-ON/Tet-OFF)[10,11,16]. Thus, directed evolution seems to be the only tractable strategy for developing useful transcriptional controllers in yeast.

Several studies have exploited the evolutionary design of yeast genetic switches using fluorescence-activated cell sorting (FACS)[17,18] or classical, colony-based selection systems[19–21]. Interestingly, however, directed evolution has not become the standard choice for the development/improvement of yeast genetic switches, although this strategy is routinely employed in the creation/evolution of prokaryotic ones[5,22–28]. In yeast genetics, a number of positive and negative selection markers have been developed. However, most of these markers are not as useful as ON- and OFF-state selectors of genetic switches, given the lack of tunability in their selection thresholds. In practice, it is very difficult to isolate the best performers among the given variant pools because experimentalists are blind to the functional distribution of the library. To make the selection work, one must systematically screen for appropriate selection conditions with minimized false-positives and -negatives.

Here, we describe an evolutionary platform for the fast-track parallel construction/evolution of yeast genetic switches using a tri-functional fusion protein, consisting of a negative selector (herpes simplex virus thymidine kinase, hsvTK[25,29,30]), a positive selector (Zeocin-resistance protein, Ble[31–35]), and a fluorescent reporter (green fluorescent protein, GFP[36]) (Fig. 1a). This fusion protein, hsvTK-Ble-GFP (TBG), provides three advantages. First, TBG permits seamless ON/OFF selection of the improved genetic switches. Second, all selection processes can be completed using liquid handling alone, and one can test multiple selection conditions in parallel using multi-well plates. Third, all of the selected variants can be characterized seamlessly by fluorescence analysis, enabling the systematic generation of yeast genetic switches with different specifications. Leveraging the robust and flexible nature of our directed evolution platform, we created various useful yeast genetic switches with improved S/N ratios, high inducer sensitivities, sensory functions, and reversed switching behaviours. The resulting genetic switches were easily integrated into biosynthetic pathways, enabling selective AND-gated control. Using the rapid and highly flexible workflow described herein, it is now possible to de novo constructing and implementing various yeast genetic switches for different purposes and in various contexts.

## Results

**Grand design of the directed evolution platform.** To establish a truly useful evolutionary platform for the development of yeast genetic switches, a selector system should possess a number of features. First, liquid-based operation is important to test the different selection conditions in parallel and to permit iterative rounds of selection with progressive elevation of selective pressure. Second, the difference in the expression level among the variants should be translated directly into the survival rate during both the ON and OFF selections, thereby facilitating the progressive enrichment of the desired variants. Third, the selection threshold should be tunable so that all genetic switches can be evolved using the same platform, irrespective of the output levels of the parent and targeted specifications. Fourth, a decent throughput/efficiency is necessary to identify rare mutants with the desired performance (e.g., high S/N ratio and high inducer sensitivity). Throughput/efficiency is especially important for evolving yeast genetic switches; our inability to predict the nucleotide positions to be altered to appropriately modulate their expression level[38–40] increases the size of the sequence to be randomized, and therefore the rarity of the mutants to be selected. Finally, because no selection systems are free from false-positives or -negatives, selected variants should be seamlessly and individually evaluated for their switching performance in a quantitative and high-throughput manner.

To meet all five of the above requirements, we designed a hybrid selector protein, TBG (Fig. 1a). The TBG-encoding gene contains the Zeocin-resistance gene (*ble*) from *Streptoalloteichus hindustanus* as the ON-state selector. This protein has been widely used as a positive selection marker for plasmid maintenance and recombineering in yeasts[31–35]. Importantly, the Ble dimer inactivates Zeocin by binding and sequestering the antibiotic at a 1:1 molar ratio. Thus, the Zeocin resistance of the cell should directly reflect the expression level of Ble[41,42]. We hypothesized that with this selection mechanism, we should be able to freely set the selection threshold of the switch output (TBG expression) simply by changing the amount of Zeocin.

Only a handful of negative selectors[43] have been exploited as OFF-state selectors of genetic switches. Among the candidates that have been successfully used in *E. coli*, we adopted a nucleoside kinase-based system (hsvTK)[44]. This enzyme is known to phosphorylate various nucleoside analogues with different kinetics; the resulting phospho-nucleosides have differing toxicities, providing opportunities to tweak the threshold of the OFF-state selection.

By fusing these selection systems with a GFP reporter for the evaluation of genetic switch performance, one can seamlessly evaluate the switching performance before and after the OFF/ON selections. We considered this evaluation an important step in robustly isolating the best-performing switches from the pool of survivors (Fig. 1b).

**Molecular breeding of selection/screening devices.** The TGB-encoding gene was placed downstream of a synP ($p_{tetO7}$), which consists of seven copies of the *E. coli tet* operator (*tetO*) fused to the *S. cerevisiae GAL1* core promoter ($p_{GAL1-c}$) (Fig. 2a). The

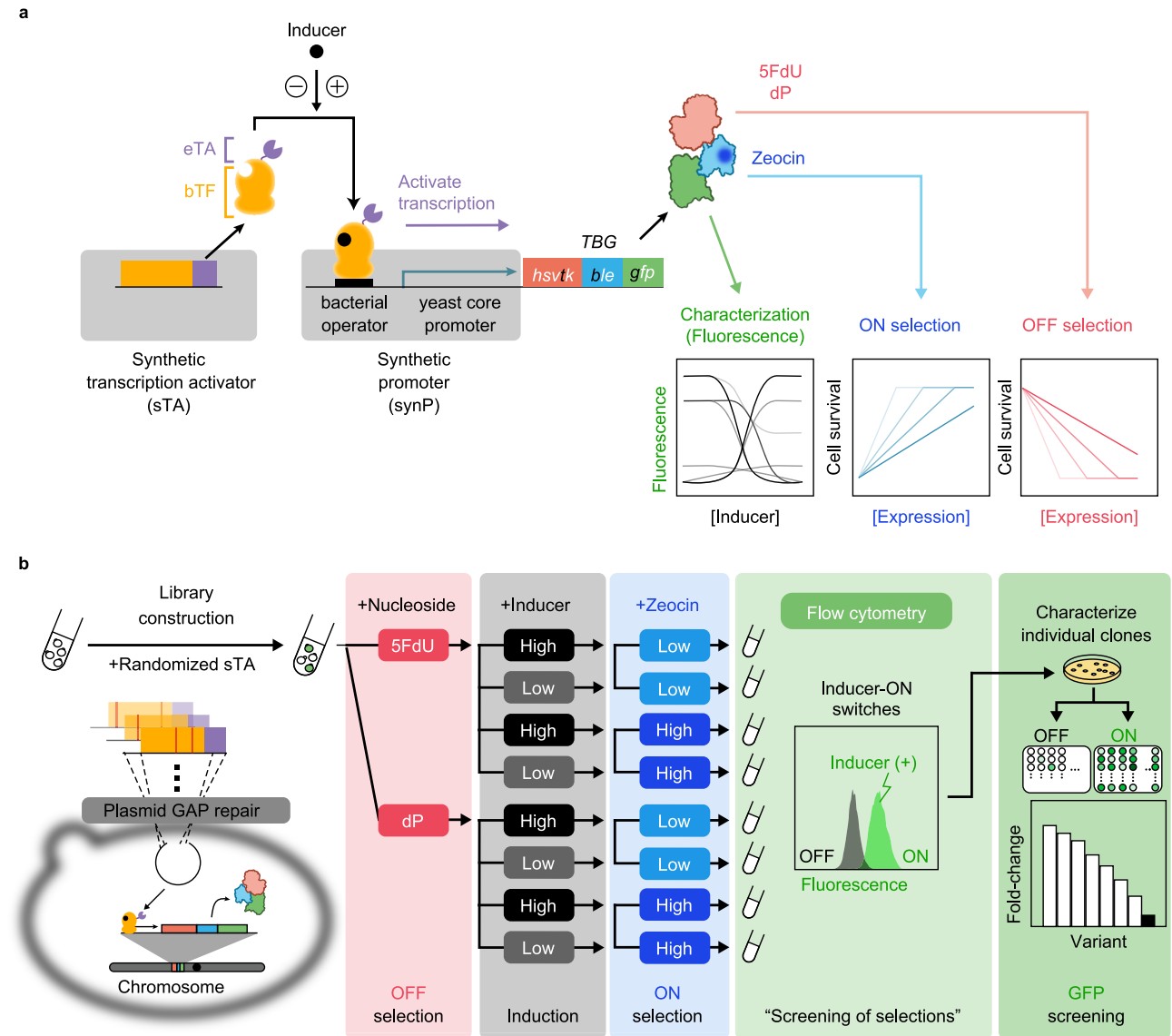

**Fig. 1 The proposed workflow for evolving yeast genetic switches. a** A synthetic promoter (synP) is constructed by fusing a core promoter with an operator sequence(s), where the synthetic transcription activator (sTA) binds in positive or negative response to an inducer to activate the transcription of a downstream gene encoding a trifunctional fusion protein (hsvTK-Ble-GFP, TBG), the genetic switch output. Then, the desired switch variants are enriched with the indicated OFF/ON selections. OFF selection: the variants with improper (leaky) switch output (hsvTK expression) phosphorylate 5-fluoro-2′-deoxyuridine (5FdU) or 6-(β-D-2-deoxyribofuranosyl)-3,4-dihydro-8H-pyrimido[4,5-c][1,2]oxazin-7-one (dP)[37] and subsequently perish owing to the toxic effect of nucleoside monophosphates (5FdUMP and dPMP). ON selection: the variants with insufficient switch output (Ble expression) are killed with Zeocin. **b** Directed evolution of yeast genetic switches is performed as follows. Yeast cells are transformed first with the genetic switch library consisting of randomly mutagenized component(s) (e.g., sTA-encoding expression cassette), followed by the OFF selection with 5FdU or dP. Next, different concentrations of inducer are added to the resultant cell pools, followed by the ON selection with varied concentrations of Zeocin. The resultant cell population is evaluated for the inducer-dependent fluorescence shift using flow cytometry to screen the populations enriched for the desired switch variants (screening of selections), from which the best-performing variants can be isolated by evaluating individual cellular GFP fluorescence in both the OFF and ON states.

TGB-encoding construct was integrated into chromosome V of the *S. cerevisiae* (for details, see Supplementary Fig. 1). We expected that rTetTA (sTA previously referred to as rtTA2$^S$-M2[45]), a fusion of the reversed-bacterial transcriptional repressor rTetR with three copies of the viral transcription activator motif (3 × VP16, VP48), would bind to $p_{tetO7}$ and enhance transcription of the TBG-encoding gene upon binding to Dox. Indeed, the cellular fluorescence of the resultant strain was responsive to Dox (Fig. 2b), indicating that TBG is useful as a reporter for promoter strength. Owing to its intrinsic leakiness, $p_{tetO7}$ had low but detectable activity even in the absence of rTetTA. When rTetTA was expressed from a plasmid (Supplementary Fig. 2), cellular fluorescence elevated by an additional fourfold, even in the absence of Dox. Because of this high OFF-state output, the addition of Dox resulted in an increase in fluorescence of only approximately threefold (Fig. 2b).

We evaluated the performance of the *TBG* construct as an ON selector. With a *TBG* incorporating the wild-type *ble* gene (i.e., $TB_{WT}G$), our original system failed to discriminate the threefold difference in promoter strength induced by Dox, with rTetTA expressed from a plasmid; the cells harbouring the $TB_{WT}G$ grew at all tested concentrations of Zeocin, indicating that the cellular

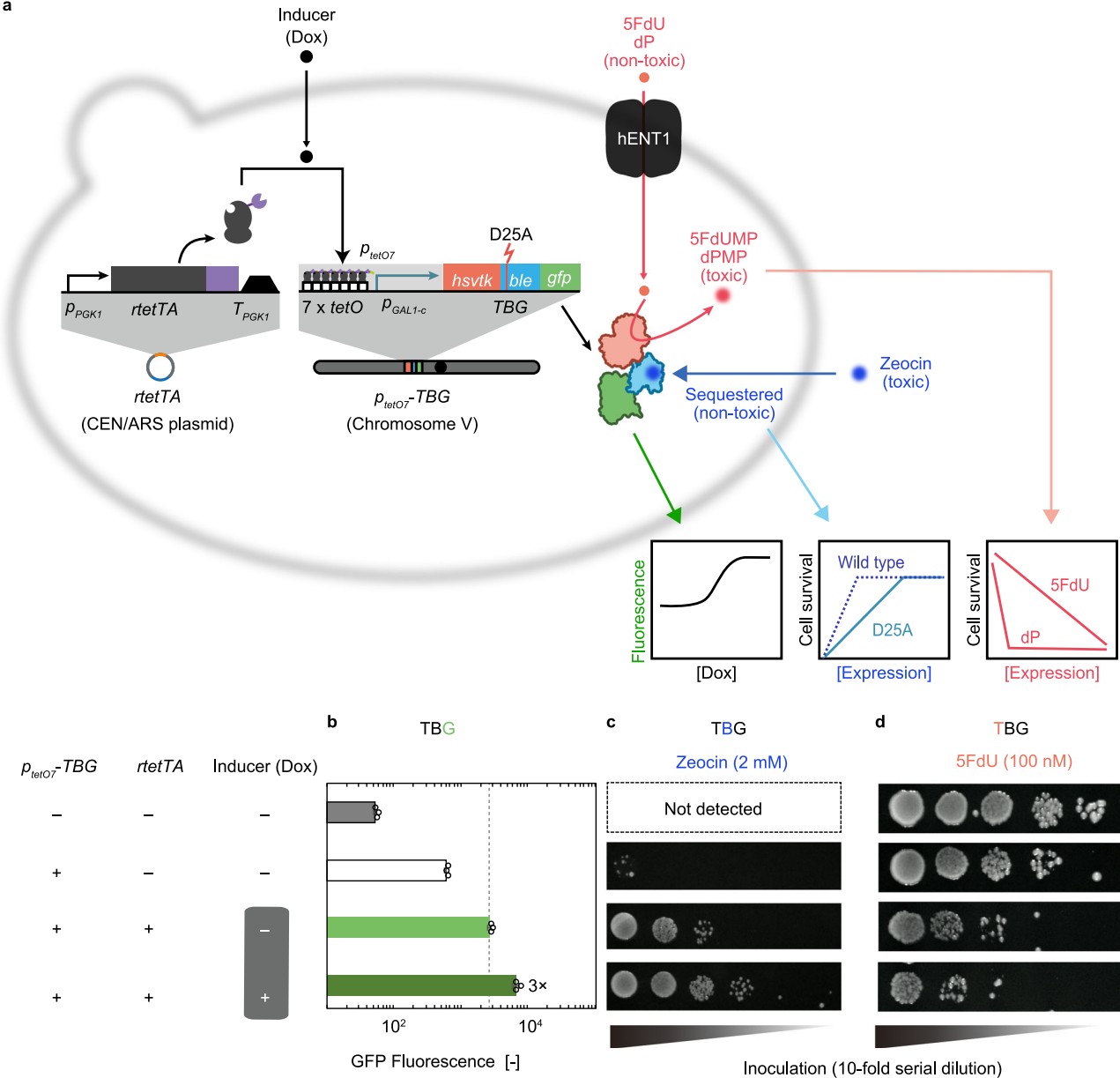

**Fig. 2 Performance of TB$_{D25A}$G as a trifunctional reporter/selector. a** Schematic illustration of Tet-ON-controlled $TB_{D25A}G$ gene expression. Nucleosides (5FdU or dP) are incorporated via hENT1 and are toxified by the kinase activity of TB$_{D25A}$G. Zeocin exerts antibiotic activity towards cells, which is alleviated by TB$_{D25A}$G expression. The expression level of TB$_{D25A}$G can be quantified by GFP fluorescence. **b–d** Functions of TB$_{D25A}$G. Chromosomal expression of $TB_{D25A}G$ under control of the Tet-ON system was measured by **b** cellular GFP fluorescence, **c** cell viability in Zeocin-containing medium, or **d** cell viability in the 5FdU-containing medium. Error bars represent the mean ± SD of three independent experiments. Source data are available in the Source Data file.

Ble activity of TB$_{WT}$G was already saturated in the absence of Dox (Supplementary Fig. 3). Therefore, we introduced a single amino-acid substitution (D25A) in Ble so that the function of this protein was slightly compromised[46]. Indeed, cells harbouring the resultant $TB_{D25A}G$ construct turned out to be superior ON selectors, and in the medium containing 2 mM Zeocin, the survival rate of the $TB_{D25A}G$-containing cells was effectively proportional to the cellular fluorescence from TBG (Fig. 2c).

Previously, hsvTK was shown to be an excellent kinase for 6-(β-D-2-deoxyribofuranosyl)-3,4-dihydro-8H-pyrimido[4,5-c][1,2] oxazin-7-one (dP) and 5-fluoro-2'-deoxyuridine (5FdU), and it has been used successfully as a negative selection marker in *E. coli*[25,29,30]. Prior to our evaluation of the *TBG* gene as an OFF selector, the gene for a broad-substrate nucleoside transporter

(human equilibrative nucleoside transporter 1 (hENT1))[47–49] was cloned and integrated into the yeast chromosome, facilitating nucleoside uptake by the cell (Supplementary Fig. 4). The resultant strain turned out to be highly sensitive to dP; introduction of $p_{tetO7}$-*TBG* completely eliminated the viability of the cell, even at very low concentrations (~50 nM) of dP, irrespecitive to the presence of Dox and/or rTetTA (Supplementary Fig. 5). Thus, dP selection should be probably suitable for selecting highly stringent genetic switches with near-zero leakiness but not for most of existing induction systems exhibit non-negligible leakiness (basal expression). In such cases, the variants should be selected using 5FdU as the selective agent that exhibited a graded (dose-dependent) effect: when 100 nM 5FdU was added to the medium, yeast harbouring rTetTA lost viability

in the presence of Dox (high mode), whereas such cells exhibited reduced but detectable viability even in the absence of Dox (low mode) (Fig. 2d). This is in good agreement with the leaky basal (uninduced) TBG fluorescence (Fig. 2c).

**Directed evolution of the signal-to-noise ratio of Tet-ON system**. The Tet-ON system exhibited only a threefold induction upon Dox addition (Fig. 2b), presumably owing to the nonspecific binding of rTetTA to *tetO*[12], and possibly to the accidental emergence of a cryptic promoter upstream of $p_{tetO7}$ (Supplementary Fig. 6). To reduce nonspecific binding to *tetO*, the entire region of the *rTetTA* expression cassette was randomly mutated by error-prone PCR (epPCR), and the resultant PCR product was directly cloned into the plasmid inside the cells harbouring $p_{tetO7}$-TBG by GAP-repair cloning[50,51] (Fig. 3a). The resultant transformants (~$10^5$ unique clones) were subjected to OFF selection by incubating in 5FdU-containing liquid medium without Dox to eliminate the variants with leaky basal (i.e., Dox-independent) TBG expression. Next, in order to remove the non-functional variants, thereby enriching for better switchers, we performed ON selection under a range of conditions; specifically, the pool of OFF-selected cells was aliquoted into 14 test tubes containing various combinations of Dox (0.01–10 μg/mL) and Zeocin (1 or 2 mM), and the samples were shaken overnight.

Each of the 14 ON-selected cell mixtures was analysed by flow cytometry (Fig. 3a). Functional switches were enriched to different extents under various selection conditions. In all cases, significant enrichment of desirable (Dox-inducible) variants was observed at all of the tested non-zero Dox concentrations (Fig. 3a). In general, the output signal under the induced condition (Dox+) was higher for the pools from Runs 8–14, where higher selective pressure was applied under the ON condition (2 mM Zeocin) than for their respective corresponding pools under the lower selective pressure (Runs 1–7; 1 mM Zeocin). However, ON-selected pools with 2 mM Zeocin were contaminated with a higher proportion of always-ON variants, which exhibited high fluorescence signals even in the absence of Dox. ON selection with 2 mM Zeocin appeared to exert toxic effects even on the positive (desirable) clones with higher output upon Dox induction, thereby unnecessarily decreasing the proportion of switching variants compared with non-switching variants with the always-ON phenotype (super-activators). In both cases, lower concentrations of Dox, especially below 0.1 μg/mL, resulted in the gradual accumulation of leaky variants in the pool. In this particular case, the most efficient enrichment of functional switches was achieved with Runs 9 and 11 (Pools #9 and #11). Because we wanted to isolate variants with both the highest induced output and the highest stringency, we chose Runs 8 and 11 for subsequent plate-based screening.

From the two selected pools, 93 clones were randomly picked and subjected to fluorescence screening in 96-well plates (Fig. 3b). A total of 48% (45/93) of the tested clones exhibited more than fivefold induction with Dox (10 μg/mL), and the best rTetTA variant (rTetTA$_{K8N, L131L}$) exhibited 8- and 11-fold induction in the presence of 0.3 and 10 μg/mL Dox, respectively (Fig. 3c and Supplementary Fig. 7).

**Reconstruction of inducer-repressible switches**. In theory, any transcription factor can be used as a component of a eukaryotic transcription switch by following the protocol for developing the Tet-ON system, but the performance of the resultant switch is unpredictable because different transcription factors vary instability, DNA-binding affinity, and how their function changes when fused to other proteins or domains to generate an sTA. Moreover, the switching behaviour of sTAs is highly

dependent on the context (expression level, promoter location, copy number, and type of strain). In practice, re-created systems must be re-evolved to ensure adequate performance in a given context.

In an initial attempt to characterize negatively regulated eukaryotic genetic switches, we reconstructed two yeast transcription switches that have been described in recent literature: the 2,4-diacetylphloroglucinol (DAPG)-repressible (DAPG-OFF)[8] switch and the D-camphor-repressible (Camphor-OFF) switch[8,9] (Fig. 4a). Specifically, DAPG- and D-camphor-responsive bacterial repressors (PhlF and CamR) were fused with a VP48 activation domain and a nuclear localization signal (NLS)[9] to generate sTAs (PhlTA and CamTA, respectively) with protein sequences identical to the published sequences (Supplementary Fig. 2). Single operators for PhlF and CamR (*phlO* and *camO*, respectively), instead of seven copies of *tetO*, were fused to $p_{GAL1-c}$ to create the synthetic promoters ($p_{phlO1}$ and $p_{camO1}$, respectively (Supplementary Fig. 1)). Although our work was, as far as possible, consistent with the described methods, our initial constructs failed to function as reported, likely owing to differences in the *cis*-regulatory elements (promoter/terminator) used to drive the expression of the sTAs. In addition, we observed high toxicity of the PhlTA-expressing plasmid. Upon transformation of the plasmid into yeast, we obtained only a few viable colonies that were insensitive to DAPG (Fig. 4b and Supplementary Fig. 8a). These isolates might be mutants carrying spontaneous, down-tuning mutations in the PhlTA-expressing plasmid that appeared to mitigate the toxic effect of over-expressing VP16[52,53]. Although the CamTA-expressing plasmid exhibited lower toxicity in yeast (Supplementary Fig. 8c), the reconstructed Camphor-OFF switch showed significant leaky expression in the presence of D-camphor (Fig. 4c).

Using epPCR, we mutagenized the entire PhlTA and CamTA expression cassettes, and the resultant libraries were subjected to ON/OFF selection (Supplementary Figs. 9a and 10) according to the workflow shown in Fig. 1b. This selection permitted the rapid isolation of functional DAPG-OFF mutants (1-2E and 1-6H), which exhibited a fourfold decrease in expression upon DAPG addition (Fig. 4b and Supplementary Fig. 9a), as well as a Camphor-OFF switch variant (1–8D) with improved responsivity to D-camphor by twofold in the S/N ratio (Fig. 4c and Supplementary Fig. 10). Notably, no obvious toxicity was observed for the isolated yeast variants (Supplementary Fig. 8). Five of the best-performing camphor-OFF switches shared a frameshift mutation (A695del) that abolished the complete NLS (Supplementary Fig. 11), whereas four of the best-performing DAPG-OFF switches retained an intact NLS but carried the same mutations in the *phlTA* expression cassette (Supplementary Figs. 8b and 9b). These observations indicated that a partial compromise in NLS function (and presumably in nuclear localization) was required for optimal CamTA activity, to ensure that there was no or less leaky TBG expression in the presence of D-camphor. In a control experiment, we detected no transcriptional activation by CamTA lacking the NLS (Supplementary Fig. 12).

**Conversion of DAPG-OFF system into DAPG-ON system**. Mutations sometimes result in the emergence of transcription factors with different switching behaviours. Various bacterial repressors have been shown to invert their ligand responses following the acquisition of a limited number of mutations[24,25,55–64]. Therefore, we tested whether mutagenesis of the gene encoding the aforementioned DAPG-OFF switch could convert the construct into a DAPG-ON switch. To this end, the same PhlTA library used for the isolation of DAPG-OFF switches was

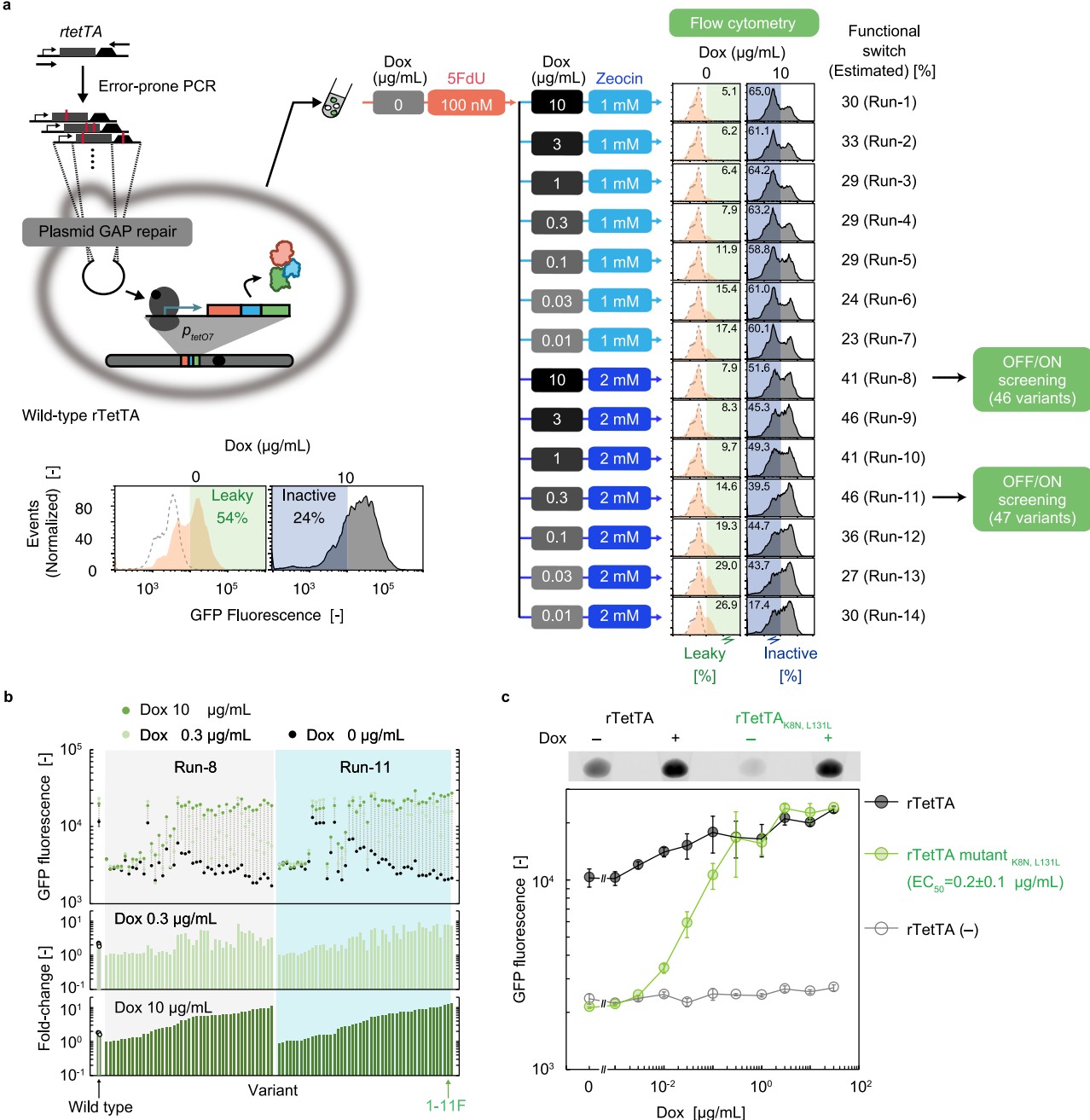

**Fig. 3 Directed evolution of the S/N ratios of Tet-ON switches. a** Random mutations were introduced into the rTetTA-encoding expression cassette of the Tet-ON switch, and the resultant library was subjected to parallel operation of OFF/ON selections under different selection conditions. The resultant cell populations were evaluated for Dox-dependent fluorescence shifts to screen for the conditions under which switch variants were enriched. The green and blue areas indicate the relative abundance (%) of (respectively) the unwanted behaviour, i.e., leaky variants (cells with GFP fluorescence more than $10^4$ in the absence of Dox) and the inactive variants (cells with GFP fluorescence >$10^4$ in the presence 10 μg/mL Dox). Relative abundance (%) of the switching variants was obtained by subtracting both of these two numbers from 100%. Dashed lines indicate the histogram obtained from yeast strains carrying plasmid without rTetTA. **b** A 96-well-plate-based fluorescence screening of the Tet-ON variants. **c** Transfer function of the selected mutant. Error bars represent the mean ± SD of three independent experiments. The 50% effective concentration ($EC_{50}$) values were calculated from the dose–response curve fitted to the Hill equation by the least-squares method. The top panel represents the fluorescence image for the cell pellets of yeast strains harbouring wild-type and evolved Tet-ON switch (rTetTA$_{K8N, L13L}$) incubated with or without 10 μg/mL Dox. Source data are available in the Source Data file.

subjected to OFF selection in the absence of DAPG followed by ON selection in the presence of this compound (5 μM). From the survivor pools, 30 variants were randomly picked and subjected to the OFF/ON screenings (Supplementary Fig. 13a), resulting in the isolation of a DAPG-ON variant (1–11E) with an eightfold DAPG-dependent increase in fluorescence (Fig. 4d and

Supplementary Fig. 14a). Mutational analysis of this variant revealed the presence of three mutations: one (Q117R) was essential for the functional reversal of PhlTA, and the other two mutations (E143K and/or K86T) were required for the improved response (both in sensitivity and fold-change) of the reversed switch (Fig. 4e and Supplementary Fig. 13b). Another cycle of

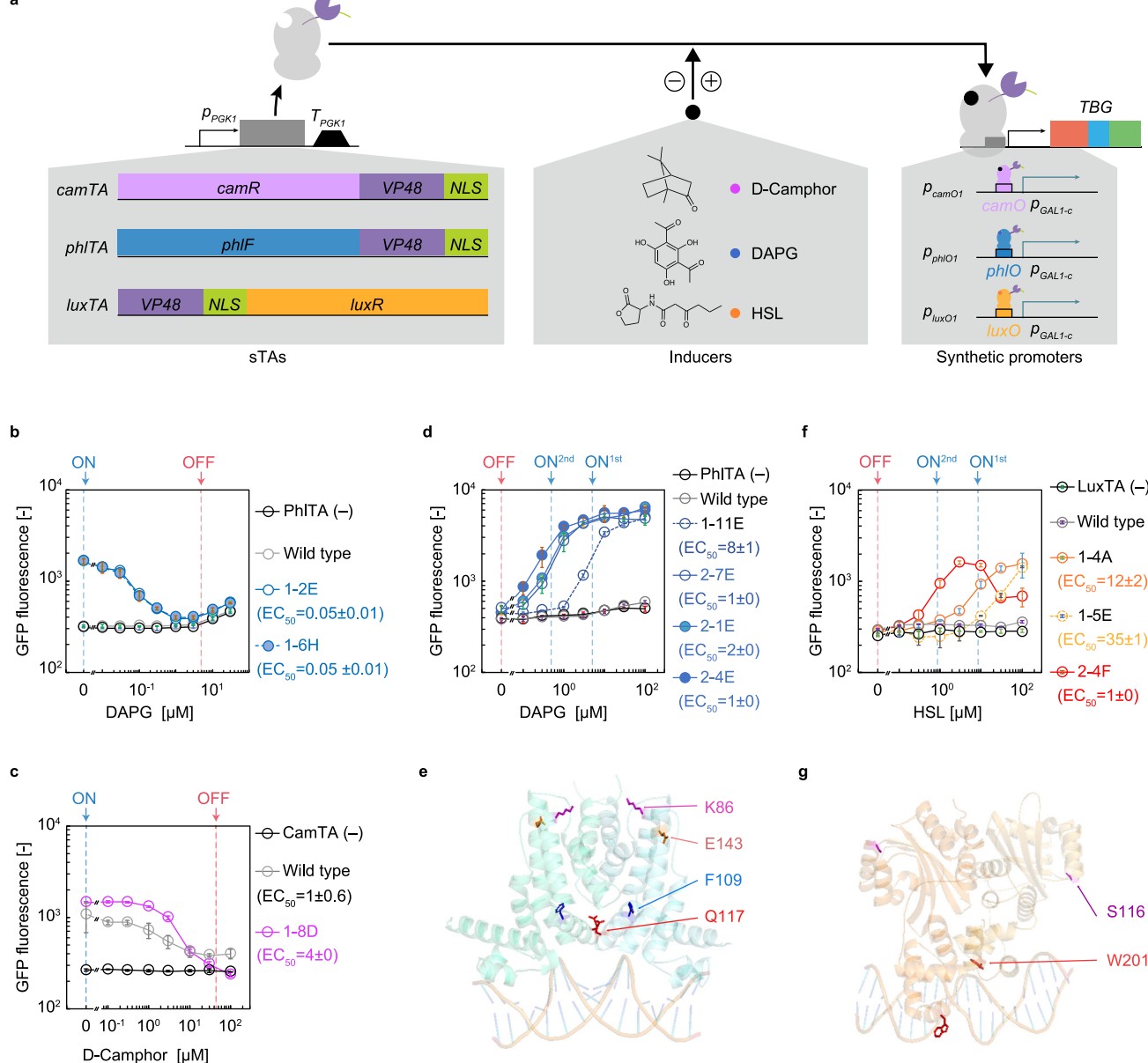

**Fig. 4 Creation and evolution (as assessed by S/N ratio, sensitivity, and behaviour type) of yeast transcriptional switches. a** Yeast genetic switches developed in this study. The plasmid expression of the identified sTA variants provided dose-dependent activation genes placed under the control of $p_{phlO1}$, $p_{camO1}$, or $p_{luxO1}$ in the host cells. The transfer function of the parent and variants of the DAPG-OFF switch (**b**), Camphor-OFF switch (**c**), DAPG-ON switch (**d**), and HSL-ON switch (**f**). TBG-derived GFP fluorescence is plotted as a function of each inducer concentration. Error bars shown represent the mean ± SD of three independent experiments. The concentrations of inducers added during the OFF/ON selections are indicated by arrows and dashed lines. The $EC_{50}$ values were calculated from the dose–response curves fitted to the Hill equation by the least-squares method and are expressed in micromolar concentration units. **e**, **g** Structural mapping of mutations in PhlF and LuxR that reversed/sensitized PhlTA, and sensitized LuxTA, respectively. The structures of PhlF and LuxR were modelled using the Swiss-Model server[54] based on the crystal structure of the TetR family transcription regulator SCO0332 (PDB: 2ZB9) and quorum-sensor protein TraR (PDB: 1L3L), respectively. The DNA structures are taken from the corresponding reference crystal structures. Source data are available in the Source Data file.

mutagenesis, selections, and screenings, using a lower concentration of DAPG (0.5 μM) for the ON selection (Supplementary Fig. 13c), yielded three second-generation variants (2-1E, 2-4E, and 2-7E) that exhibited $EC_{50}$ values that were decreased by up to eightfold (Fig. 4d). All of these second-generation variants shared the same mutation (F109L) (Fig. 4e and Supplementary Fig. 14b), which was subsequently found to be fully responsible for the increased sensitivity to DAPG (Supplementary Fig. 13d). PhlF has previously been engineered for its sensitivity and selectivity to DAPG[5] in *E. coli*, but the mutations that provided an inverted

PhlF (rPhlF) function in the present study have not been reported previously (to our knowledge). Similarly, mutations that sensitize rPhlF have not been reported.

**Evolving a yeast quorum sensor**. To date, no prokaryotic transcriptional activators have been exploited as components of yeast sTAs. Therefore, we sought to generate sTAs using *Vibrio fischeri* LuxR, a 3-oxo-hexanoyl homoserine lactone-induced bacterial transcriptional activator as the sensory component. Simple

plasmid expression of LuxTA (a fusion of LuxR, VP48, and NLS, Fig. 4a) did not confer *N*-(ketocaproyl)-D,L-homoserine lactone (HSL)-dependent expression of TBG under the control of $p_{luxO1}$ (the *GAL1* core promoter fused with a LuxR-binding box [*luxO*]) (Fig. 4f). Starting from this nonfunctional parent, we conducted two rounds of mutagenesis and selection with stepwise decreases in inducer concentration (first round, 10 μM HSL (Supplementary Fig. 15a); second round, 1 μM HSL (Supplementary Fig. 15b)), successfully identifying functional HSL-ON variants. Thus, we quickly enabled yeast to sense and respond to bacterial quorum signals.

The best-performing variant, which exhibited sixfold enhancement of TBG expression upon addition of 3 μM HSL (Fig. 4f), carried two nonsynonymous mutations (S116Y, W201R) (Fig. 4g and Supplementary Fig. 16). In the context of the yeast HSL-ON switch, either of the mutations was sufficient to sensitize LuxTA to HSL alone, although to different extents (Supplementary Fig. 15c). Interestingly, however, each of the mutations behaved quite differently in *E. coli* (Supplementary Fig. 15d). As seen in yeast, the S116Y mutation, which is located in the ligand-binding region of the LuxR structure, increased the sensitivity of LuxR/$p_{lux}$ transcription controllers in *E. coli*. In contrast, the other mutation (W201R), which is located on the surface of LuxR, is known to interact with *E. coli* RNA polymerase[65], significantly compromised expression from the *lux* promoter in *E. coli*. We infer that the W201R substitution negatively impacts LuxR's function as a recruiter of RNA polymerase, which might explain how this mutation had remained undiscovered in the extensive screening programs[26,66,67] that sought LuxR mutations that sensitize the protein for activation by HSL. In the context of yeast HSL-ON switches, the role of LuxR is simply to facilitate HSL-induced DNA binding; in eukaryotic cells, the ability to capture bacterial RNA polymerase is no longer required. Thus, we postulate that this sensitizing mutation could be identified only in the context of yeast genetic switches.

**Integration of yeast switch variants into AND-gated β-carotene biosynthetic pathways.** Having developed a series of yeast genetic switches with improved S/N ratios, we sought to apply these constructs for pathway flux control. Three of the sTAs isolated in this study, DAPG-ON$_{2-1E}$, HSL-ON$_{2-4F}$, and Tet-ON$_{K8N, L131L}$, were integrated into different chromosomes in yeast (Fig. 5a). The resultant strain was transformed with plasmids on which the *gfp* gene was cloned downstream of the respective synPs ($p_{tetO7}$, $p_{phlO6}$, and $p_{luxO5}$). Note that the putative cryptic promoter sequence in the vector sequence >500 upstream of the synP (Supplementary Fig. 6) was omitted, and the number of repeats of *phlO* and *luxO* increased. The GFP fluorescence of the resulting cells was induced only in the presence of their cognate inducers, with induction by factors of >$10^2$ (Fig. 5b). Each of these synthetic promoters was stringent; that is, they exhibited low basal TBG expression in the absence of an inducer.

To leverage their orthogonal regulation capacity, we attempted to use these switches as regulators of the β-carotene biosynthesis pathway. For this purpose, the synthesis of β-carotene was performed as a nine-step process conducted by the action of three enzymes, employing *S. cerevisiae* Bts1p in combination with the *Xanthophyllomyces dendrorhous* CrtYB and CrtI proteins (Fig. 5c).

When yeast was transformed with plasmids expressing all three genes (*BTS1*, *crtYB*, and *crtI*) under the control of constitutive promoters, the yeast produced β-carotene irrespective of the presence or absence of inducers (Fig. 5d). We replaced the promoters for *BTS1* and *crtYB* with $p_{phlO6}$ and $p_{tetO7}$, respectively. Because all three of these genes needed to be

expressed, β-carotene production was expected only in the presence of both DAPG and Dox (AND-gated control). However, the promoter leakiness of $p_{tetO7}$ turned out to be non-negligible in this context, leading to significant mis-pigmentation in the absence of Dox. This leakiness was alleviated by converting the genetic switch used to regulate *crtYB* from $p_{tetO7}$ to $p_{luxO5}$, placing the β-carotene biosynthesis pathway under AND-gated control by the combination of HSL and DAPG. In this strain, we observed the expected AND-gate behaviour where β-carotene production was observed only in the presence of both DAPG and HSL. A similar strategy was employed in the construction of a strain in which the β-carotene biosynthetic pathway was placed under the AND-gate control of Dox and HSL. Thus, by implementing specific combinations of the switches obtained in the present study in a plug-and-play manner, we were able to quickly construct an AND-gate-controlled heterologous pathway in yeast.

**Discussion**

Classical agar plate-based ON and OFF selection systems, such as His3p/3-aminotriazole (3-AT) and Ura3p/5-fluoroorotic acid (5-FOA), fail to discriminate between improved and prototypic genetic switches when their output levels under selection are below or above the given selection threshold[19,20]. This challenge requires researchers to reconstruct the expression levels of selectors in individual contexts and/or at each round of evolutionary cycling. Even with the FACS-based ON and OFF selections where the selection threshold can be easily modulated by changing the sorting threshold[17,18], experimentalists are required to repeat the sorting experiments until the correct gating condition is achieved. Instead of repeating, for instance, the multiple FACS-sorting experiments, our selection platform allows us to operate the multiple selections in a multi-well format, with different selection conditions. Given that the processes described here can be implemented by liquid-based operation, our workflow can be fully automated and can be combined with recently developed in vivo mutagenesis technologies[68,69] to further accelerate the evolutionary cycle of yeast genetic switches.

Using the workflow described in Figs. 1 and 3a, we isolated multiple variants with inverted switching behaviours (DAPG-ONs, Fig. 4d), HSL sensors (HSL-ONs, Fig. 4f), and variants of the HSL sensors with improved HSL sensitivities (Fig. 4f). The genetic switches described here might need to be re-optimized in other contexts because of the intrinsic context-dependency[70], it should be possible to quickly re-adapt them to different contexts as was described in this paper. Indeed, one of the best-engineered sTAs, rTetTA, first malfunctioned in our system, exhibiting low stringency (Fig. 2). Similarly, two other previously reported sTAs, PhlTA, and CamTA, were virtually non-functional in the context tested in the present study, and one of these published switches (PhlTA) exhibited unexpected and severe cell toxicity (Supplementary Fig. 8). Nonetheless, a single round of randomization and OFF/ON selections rapidly recovered the reported behaviour in a different context.

Unlike the mock selection, where the functional genetic switches outcompete non-functional (always-ON and -OFF) variants, the actual genetic switch library consists of variants with different characteristics. In most cases, libraries are dominated by wild-type-like (weakly inducible) variants, and it is very important to find the appropriate selection condition that can selectively enrich rare mutants with slightly improved performance. To test whether our selection platform can be applied to real-world applications, we selected an improved Tet-ON switch variant (Tet-ON$_{K8N, L131L}$) against an ~1:2000 background (with wild-type rTetTA or empty vector). Surprisingly, despite the limited difference in the performance of these variants, only a

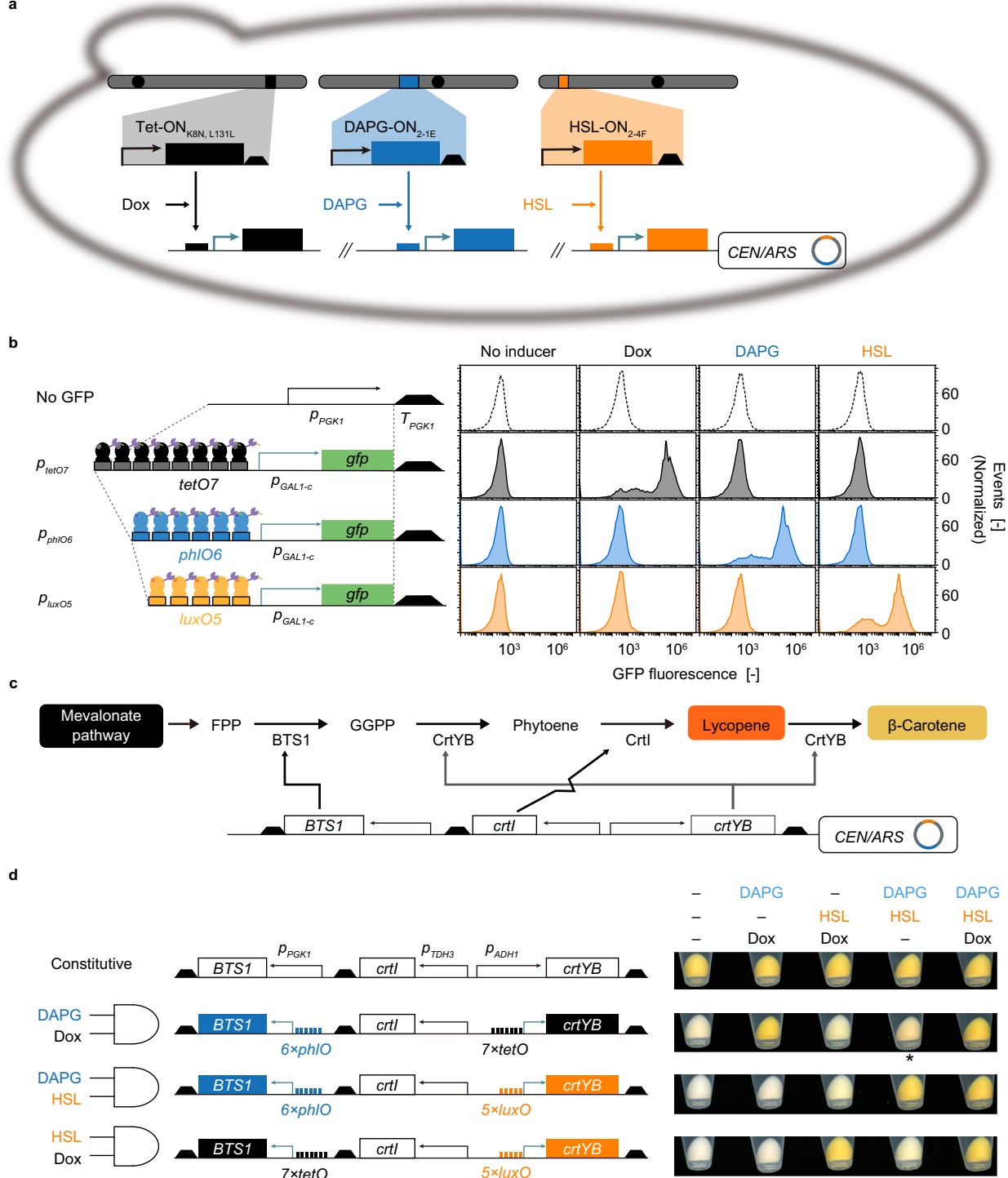

**Fig. 5 Regulation of the flux from FPP toward β-carotene by using the developed switches. a** Yeast strain used for the AND-gated control of carotenoid biosynthesis. The three plasmids expressing sTAs for use by the DAPG-ON$_{2\text{-}1E}$, HSL-ON$_{2\text{-}4F}$, and Tet-ON$_{K8N, L131L}$ switches were chromosomally integrated. The gene downstream of each synP ($p_{phlO6}$, $p_{tetO7}$, and $p_{luxO5}$) was expressed only in the presence of corresponding inducer (DAPG, Dox, and HSL, respectively). **b** Orthogonal GFP expression control using Dox, DAPG, and HSL was measured by flow cytometry. **c** The pathway to β-carotene. *FPP* farnesyl diphosphate, *GGPP* geranylgeranyl diphosphate. **d** Constitutive and AND-gate-controlled β-carotene biosynthesis. Cell pellets of yeast strains (right panel) expressing (*BTS1*, *crtYB*, and *crtI*) under the control of constitutive promoter or synP ($p_{phlO6}$, $p_{tetO7}$, and $p_{luxO5}$) in the combinations shown in left panel. These strains were inoculated into liquid medium with Inducers (Dox, DAPG, and HSL) in different combinations and incubated at 30°C for 24 hours. The concentrations of the inducers were as follows: DAPG (3 μM), HSL (3 μM), and Dox (10 μg/mL).

single round of OFF/ON selection yielded a 133-fold enrichment of Tet-ON$_{K8N, L131L}$ over the background (Supplementary Figure 17). This result highlights the robustness of the platform.

A possible drawback of our selection platform is the requirement of two genes for OFF selection. In theory, loss-of-function mutations could occur both in *hsvtk* or *hENT1* and they would lead to the selection escapes. Indeed, we observed both 5FdU- and dP-resistant clones with a frequency of $10^{-3}$ (Supplementary Table 1), which were higher than those reported for other systems[71,72]. To reduce this mutation-based selection escape, duplication of negative selector genes is effective[73]. The chromosomal addition of another *hENT1* expression cassette resulted in a >100-fold reduction of escapees (Supplementary Table 1). This indicates that the major source of selection escapees emerges via mutations at *hENT1*, not *hsvtk*. It is likely that the hENT1 expression is toxic to yeast, thereby imposing selective pressure on its loss.

The yeast promoters are known to be affected by their surrounding sequences[74], and the level of leakiness of synPs appears to be quite different depending on the reporter type (phenotype). Indeed, we observed the leakiness of the Tet-ON$_{K8N, L131L}$ switch when it was applied to control the carotenoid biosynthesis gene crtYB (Fig. 5D), whereas no detectable leakiness was observed for GFP expression (Fig. 5b). This could be simply because even the least expression of CrtYB results in pigment biosynthesis, or it could be from a transcription read-through from the upstream sequence. Evidently, adding this sequence to $p_{tetO7}$ caused significant leakage of the GFP expression (Supplementary Figure 18). A set of terminators and ribozyme sequences could be used to insulate synPs from their upstream sequences to block their transcription read-through[75].

Given the shortage of reliable tools, yeast synthetic biology has provided limited capacity for metabolic control. Using our platform, we created dozens of inducible genetic switches with distinct specifications. We tested whether these switches could be independently used to control pathway genes in one cell (Fig. 5d). Among them, the performance of the DAPG-ON switch was comparable to that of the Tet-ON switch, which is known as one of the best switches in terms of its fold induction (>10$^2$) (Fig. 5b), as well as that of the recently reported high-performing switches[75]. We then demonstrated that these evolved switches could be further improved in fold induction simply by tweaking the numbers of operators[10] tandemly aligned in synPs. Specifically, increasing the number of *luxO* repeats from 5 to 10 ($p_{luxO10}$ instead of $p_{luxO5}$) greatly improved the fold change of the HSL-ON switch to levels comparable to those obtained with both DAPG-ON and Tet-ON switches (Supplementary Fig. 19). In summary, our workflow enables the in situ construction of high-quality genetic switches that are useful in various synthetic biology applications.

## Methods

**Materials**. dP was purchased from Berry & Associates (Catalogue # PY7270; Dexter, MI, USA); stock solutions were prepared by dissolving appropriate amounts of dP in dimethyl sulfoxide to generate a 1000× stock (stored at 4 ℃). Zeocin$^{TM}$, 5FdU, Dox, HSL, DAPG, and D-camphor were purchased from InvivoGen (Catalogue # ant-zn-1 and ant-zn-1p; San Diego, CA, USA), Tokyo Chemical Industry (Catalogue # D2235; Tokyo, Japan), Clontech Laboratories (Catalogue # 631311; Mountain View, CA, USA), Carbosynth (Catalogue # FK29472; San Diego, CA, USA), Santa Cruz Biotechnology (Catalogue # Sc-206518; Dallas, TX, USA), and Nacalai Tesque (Catalogue # 07007–62; Kyoto, Japan), respectively. The oligonucleotides used in this study were synthesized by Eurofins MWG Operon (Ebersberg, Germany). All other chemicals and media had the highest available grade.

**Strains, plasmids, and media**. The strains used in this study are listed in Supplementary Table 2. *S. cerevisiae* BY4741 was used as the parental strain in this study. Yeast cells were grown in YPD (1% yeast extract, 2% peptone, and 2%

D-glucose) or SD (0.67% yeast nitrogen base without amino acids (BD Diagnostic Systems, Sparks, MD, USA) and 2% D-glucose) medium. Amino acids and nucleotides (60 mg/L leucine, 20 mg/L histidine, 20 mg/L methionine, and 20 mg/L uracil) were added to the SD medium as required by the auxotrophic strains. Agar (1.5%) was added to the solid medium. The plasmids and primers used in this study are listed in Supplementary Table 3 and Supplementary Data 1, respectively. All sequences used in this study are listed in Supplementary Table 4. The plasmids pRS405red, pRS406red, pRS403red, and pATP403red were derived from pRS405 (ATCC, Manassas, VA, USA), pRS406 (ATCC), pRS403, and pATP403, respectively. To construct pRS403red, pRS405red, and pRS406red, the *HIS3*, *LEU2*, and *URA3* markers were replaced with extended derivatives (*HIS3red*, *LEU2red*, and *URA3red*, respectively) to improve the plasmid integration efficiency. pFS181[76] was a gift from Nick Rhind (Addgene plasmid # 12536) for use in the PCR amplification of *hENT1*. The amplicon was cloned into pATP403red such that *hENT1* was flanked upstream by the *TDH3* promoter ($p_{TDH3}$) and downstream by the *TDH3* terminator ($T_{TDH3}$), and the resulting plasmid was designated pATP403red-*hENT1* and integrated into yeast strain BY4741, yielding BY4741-hENT1.

To construct the sTA expression plasmids (Supplementary Fig. 2), the sTA-encoding genes were cloned into pGK415 using the SalI and BglII sites between the *PGK1* promoter ($p_{PGK1}$) and terminator ($T_{PGK1}$). sTA-encoding genes were synthesized as codon-optimized open-reading frames by GeneArt® Strings DNA fragments (Thermo Fisher Scientific, Carlsbad, CA, USA), with the exception of *TetTA*, which was synthesized as a coding sequence identical to that of the Tet-ON® Advanced Inducible Gene Expression System (Clontech Laboratories). To construct the TBG-expressing plasmids (Supplementary Fig. 2), the codon-optimized *hsvtk* and *ble* genes were synthesized by GeneArt®, and the codon-optimized gene encoding monomeric umikinoko-green was amplified from pGK416-ymUkG1. The seven *tetO* repeats and $p_{GAL1-c}$ were derived from pTRE-tight (Clontech Laboratories) and pESC-URA (Agilent Technologies, Santa Clara, CA, USA), respectively. These fragments were assembled into pRS406red together with a downstream *CYC1* terminator ($T_{CYC1}$) from the pBT3-C plasmid (MoBiTec, Göttingen, Germany). The *tetO* repeat was then replaced with a single sTA-binding site (Supplementary Fig. 2). The resultant plasmids were integrated into BY4741-hENT1 to perform directed evolution experiments. All directed evolution experiments were performed with these strains harbouring a single hENT1 expression cassette. SynPs with multiple *phlO* and *luxO* target sequences were constructed via DNA synthesis and/or PCR amplification, resulting in $p_{phlO6}$, $p_{luxO5}$, and $p_{luxO10}$ (Supplementary Tables 4 and 5). These synPs, together with $p_{tetO7}$, were cloned into pGK416-ymUkG1 using NsiI and NheI sites, omitting the putative cryptic promoter sequence located in the vector sequence >500 bp upstream of the synP (Supplementary Fig. 6). To construct the plasmid encoding *crtYB*, *crtI*, and *BTS1*, these genes were cloned into pATP426 vector using FseI/AvrII, SalI/NotI, and SmaI. Then, the three-gene expression cassette was cloned into pRS416 (ATCC), and the resulting plasmid was designated pATP416-crtYBI (Supplementary Fig. 20). The *PGK1 and ADH1* promoters ($p_{PGK1}$ and $p_{ADH1}$) in the resultant plasmid were replaced with $p_{phlO6}$, $p_{tetO7}$, and $p_{luxO5}$ (in various combinations) to obtain pATP416-$p_{tetO7}$-$p_{phlO6}$-crtYBI, pATP416-$p_{luxO5}$-$p_{phlO6}$-crtYBI, or pATP416-$p_{luxO5}$-$p_{tetO7}$-crtYBI. For the co-expression of sTAs, the expression cassettes encoding the PhlTA$_{2-1E}$, LuxTA$_{2-4F}$, and TetTA$_{K8N, L131L}$ proteins were cloned into pRS405red and pRS403red, respectively, using PvuII sites. The *LEU2* marker in the plasmid carrying the *rtetTA*$_{K8N, L131L}$ gene was replaced with the *MET15red* marker. These plasmids were sequentially integrated into strain BY4741 to obtain strains BY4741-*Phl*$_{2-1E}$, BY4741-*Phl*$_{2-1E}$-*Lux*$_{2-4F}$, and BY4741-*Phl*$_{2-1E}$-*Lux*$_{2-4F}$-*Tet*$_{K8N, L131L}$ (Supplementary Fig. 21). To construct the strains harbouring two hENT1 expression cassettes, the pGK401red-*hENT1* plasmid was constructed by replacing the *rtetTA*$_{K8N, L131L}$ gene of pGK401red-*rtetTA*$_{K8N, L131L}$ with the PCR-amplified *hENT1* gene, such that *hENT1* was flanked upstream by $p_{PGK1}$ and downstream by $T_{PGK1}$. The resultant plasmid was integrated into the yeast strain BY4741-hENT1-$p_{tetO7}$-TBG, yielding BY4741-hENT1-$p_{tetO7}$-TBG/hENT1.

**OFF and ON selections**. Approximately $10^6$ cells obtained from an overnight culture were diluted in 2 mL of SD medium containing synthetic nucleosides (dP or 5FdU) or Zeocin$^{TM}$ (initial optical density at 600 nm (OD$_{600}$) ~0.1) and shaken for 15 h at 30 ℃. As a reference culture, another aliquot of $10^6$ cells (OD$_{600}$ ~0.1) was shaken in the same medium without nucleosides or Zeocin$^{TM}$. Cells in the resulting cultures were pelleted, washed with sterile distilled water (DW), resuspended in sterile DW, serially diluted, and spotted onto SD plates with appropriate amino acids to evaluate cell viability. Pictures of the resulting colonies were recorded after 2 (ON selection) or 3 days (OFF selection) of growth at 30 ℃. Zeocin concentrations used were 1.6, 3.2, or 4.8 mg/mL (corresponding to 1, 2, or 3 mM). 5FdU and dP were used at concentrations of 50, 100, or 1000 nM.

**Characterization of genetic switches (fluorescence measurements)**. Typically, a 5-μL aliquot of the transformant culture was added to 495 μL of fresh SD medium in 96-deep-well plate and incubated with shaking for 24 h (30 ℃, 1000 rpm) with different concentrations of inducers. For the functional screening of Tet-ON, DAPG-OFF switches, and the first-generation DAPG-ON and HSL-ON switches, the GFP fluorescence intensity was measured by flow cytometry. In brief, the

collected cells were suspended in DW or sheath solution, and the GFP fluorescence of at least 10,000 cells was measured with a blue (488-nm) laser and a 530/30 nm bandpass filter using a BD FACSCanto™ II flow cytometer (BD Biosciences, San Jose, CA, USA), or CytoFLEX flow cytometer (Beckman Coulter, Brea, CA, USA). BD FACS Diva software (v5 and v8) and Beckman Coulter CytExpert (v2) were used to collect all of the flow cytometry data. The median GFP fluorescence for each population was calculated using BD FlowJo™ software v.10 (BD Biosciences). To characterize the switching function, the data for each switch were fitted to the Hill equation[5]. For the functional screening of Camphor-OFF switches and second-generation DAPG-ON, HSL-ON switches, the GFP fluorescence intensity was measured using a plate reader. The cells were incubated with shaking (30 °C, 1000 rpm) for 24 h in the absence or presence of inducers, and fluorescence (ex/em = 485/510 nm) was measured using a 96-well PerkinElmer EnVision® 2104 Multi-label fluorescence reader (Waltham, MA, USA).

**Library construction**. The sTA expression cassette was amplified via error-prone PCR[77] using flanking primers (5′-gcg ggc ctc ttc gct att ac-3′ and 5′-aac cgt att acc gcc ttt gag tg-3′) and *TaKaRa Taq*™ in PCR mixtures supplemented with 50 μM MnCl₂. The PCR-amplified DNA fragments and SalI- and BglII-double-digested pGK415 vector were gel-purified and used for GAP-repair cloning[50,51] together with an overnight yeast cell culture. Transformant cultures were transferred into a selective liquid medium and incubated for more than 1 d for further operation and glycerol stock preparation. A portion of the transformants was plated onto a selective solid medium to determine cloning efficiency, from which the library size of the plasmid was evaluated. Typically, $10^3$–$10^5$ transformants were obtained. The transformant library was then directly or indirectly subjected to OFF/ON selection (Supplementary Table 6).

**Selection of genetic switches from the library**. For the isolation of switches activated by inducers (Tet-ON, DAPG-ON, and HSL-ON), a yeast library culture was diluted into SD medium containing 5FdU or dP and incubated with shaking for 24 hours (30 °C, 150 rpm, initial OD₆₀₀ ~0.1). Next, the cells were washed once with sterile DW, diluted in SD medium containing inducers, and incubated with shaking for 24 h (30 °C, 150 rpm, initial OD₆₀₀ ~0.1). Finally, the cells were transferred into SD medium containing inducers and Zeocin™ and incubated with shaking for 24 h (30 °C, 150 rpm, initial OD₆₀₀ ~0.1). The resultant cell culture was rinsed with sterile DW and spread on SD plates with appropriate amino acids to recover survivor clones for further screening. The remaining culture was used for the glycerol stock preparation. For the isolation of switches repressed in the presence of inducers (DAPG-OFF and Camphor-OFF), a yeast library culture was diluted into SD medium containing Zeocin™ and incubated with shaking for 24 hours (30 °C, 150 rpm, initial OD₆₀₀ ~0.1). Next, the cells were washed once with sterile DW, diluted in SD medium containing inducers, and incubated with shaking for 24 h (30 °C, 150 rpm, initial OD₆₀₀ ~0.1). Finally, the cells were transferred into SD medium containing inducers and 5FdU or dP and incubated with shaking for 24 h (30 °C, 150 rpm, initial OD₆₀₀ ~0.1). The resultant cell culture was rinsed with sterile DW and spread on SD plates with appropriate amino acids to recover survivor clones for further screening. The remaining culture was used for the glycerol stock preparation.

**Carotenoid pigmentation**. The plasmids pATP416-*crtYBI*, pATP416-$p_{luxO3}$-$p_{phlO6}$-*crtYBI*, pATP416-$p_{luxO3}$-$p_{tetO7}$-*crtYBI*, and pATP416-$p_{tetO7}$-$p_{phlO6}$-*crtYBI* were used to transform the yeast strain BY4741-$Phl_{2-1E}$-$Lux_{2-4F}$-$Tet_{NL}$. The resultant transformants were incubated with shaking for 24 h (30 °C, 150 rpm) with different combinations of inducers (DAPG, HSL, DAPG, and Dox). The cells were then collected by centrifugation, and photos of the cell pellets were recorded using a scanner.

**Reporting summary**. Further information on research design is available in the Nature Research Reporting Summary linked to this article.

## Data availability
The source data underlying Figs. 2b, 3, 4b–d, and 4f and Supplementary Figs. 9a, 10, 12, 13, 15, and 17–19 are provided as a source data file. PDB files used in this study is available in Protein Data Bank (PDB) (ID: 2ZB9 and 1L3L). The following plasmid can be obtained from Addgene: pGK401red-*rtetTA*$_{K8N, L131L}$ (plasmid# 165968); pGK415-*rtetTA*$_{K8N, L131L}$ (plasmid# 165969); pGK415-*phlTA*$_{2-1E}$ (plasmid# 165970); pGK415-*luxTA*$_{2-4F}$ (plasmid# 165971); pATP416-$p_{luxO5}$-$p_{tetO7}$-*crtYBI* (plasmid# 165975), pATP416-$p_{tetO7}$-$p_{phlO6}$-*crtYBI* (plasmid# 165976); pATP416-$p_{luxO5}$-$p_{phlO6}$-*crtYBI* (plasmid# 165977); pATP403red-*hENT1* (plasmid# 166027); pGK401red-*hENT1* (plasmid# 166028); pRS405red-*phlTA*$_{2-1E}$ (plasmid# 166029), pRS403red_*luxTA*$_{2-4F}$ (plasmid# 166030). All other relevant data are available from the authors upon reasonable request. Source data are provided with this paper.

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

## Acknowledgements

We thank Yuko Kawasaki and Aki Ichimichi for their technical assistance. This work was supported by JSPS KAKENHI (grant number 18K14374, 15H04189, 15K14228, 16H06450), AMED (Grant numbers JP19ae0101055 and JP19ae0101060; Project Focused on Developing Key Technology for Discovering and Manufacturing Drugs for Next-Generation Treatment and Diagnosis), NEDO (Project Number P16009; Development of Production Techniques for Highly Functional Biomaterials Using Smart Cells of Plants and Other Organisms; Smart Cell Project), and the JST-Mirai Program (Grant Number JPMJMI17EJ) from Japan.

## Author contributions

Conceived and designed the experiments: M.T., K.N., D.U., J.I., and A.K. Performed the experiments: M.T. and K.N. Analysed the data: M.T., K.N., and J.I. Wrote the paper: M.T., D.U., and J.I.

## Competing interests

The authors declare no competing interests.
