## [Peer Review File · Nature Communications]

Reviewers' Comments:

Reviewer #1:

Remarks to the Author:

Synthetic biology's increasingly complex genetic circuitry requires orthogonal genetic switches. The limited repertoire of suitable, well-characterized switches is still a major bottleneck in the field, particularly in eukaryotic systems. Tominaga et al. address this problem by having developed a selection system for evolving switchable transcription factors in yeast. Selecting for switchable function is a considerably more challenging problem than e.g. the selection of a constitutive binder: Selections for both the active (ON) and the inactive (OFF) state have to be performed and balanced in stringency to enrich genetic switches from a background of constitutively active or inactive variants. The presented selection system achieves this dual selection with a trifunctional fusion protein consisting of a positive (Ble) and a negative selector (hsvTK) protein, and a GFP to facilitate easy characterization of selected variants. The usefulness of the method is demonstrated by a variety of different examples of directed evolution for switchable transcription factors; the authors managed to substantially improve the switching behavior of two synthetic transcription factors, inverted the switching logic of one of them, and engineered a LuxR based yeast genetic switch. Evolved switches were used to construct a carotenoid synthesis pathway with AND control. This article is well-written and the presented research of notable interest to the field of eukaryotic synthetic biology, as a comparable dual selection system does not yet exist in yeast. The variety of presented use cases shows the general robustness of the system. For the reader interested in using the selection system, more quantitative characterizations would be useful to guide the design of selection experiments. Specifically, I would suggest doing mock selections of switchers from backgrounds of non-switchers to determine enrichment factors with typical selection parameters. Further, the system uses a negative selection dependent on the function of two genes (hsvTK and hENT1), i.e. a loss of function of either of these should lead to false positives. The frequency of spontaneous occurrence of escapers should be assessed, and possibly the mode of escape analyzed. In addition, it might be helpful to have a supplementary table of the actual library sizes of individual selection experiments since they varied by two orders of magnitude. This would allow conclusions about the expected frequency of desired phenotypes in random libraries and could provide researchers attempting similar selections valuable guidance as to how large libraries should be. Notably, for several selections small library sizes (10³) apparently were sufficient.

In the discussion regarding FACS-based selection it is said that "the appropriate selection threshold for a given experiment is virtually impossible to guess a priori. Lacking the ability to predict what type of mutants will emerge or at what probabilities, experimentalists are required to repeat the sorting experiments until the right gating condition happens to be achieved." This reasoning of the benefits of the presented selection system over FACS-based selections does not seem compelling, because i) appropriate controls with known activation strength (e.g. low, medium, strong) can be used to inform the FACS experimentalist of appropriate gating conditions and ii) the presented system faces the same issue, which the authors remedy by running parallel selections with different selection stringencies. This is perfectly doable with FACS-based selection as well. There are other downsides of FACS-based selections which could make for more convincing arguments, e.g. high signal scatter for low fluorescence signals causing difficulties during "OFF" selection.

Further, I am somewhat skeptical of the claim that the system "can be combined with recently developed in vivo mutagenesis technologies to conduct continuous evolution". If mutagenesis is ongoing during both ON and OFF selections it is to be expected that this does not facilitate enrichment of switchers, but spontaneous emergence of variants with high fitness in the respective environment during each selection cycle. Alternatively, mutagenesis could be performed only before dual selection, but this effectively equates to 'conventional' library generation and is not 'continuous evolution' anymore. However, if the authors have a promising strategy to employ their system in a continuous evolution approach with in vivo mutagenesis, they should briefly outline it here.

I have further specific comments on the manuscript:

Line 65: It is somewhat unclear what the authors mean by 'continuous selection'. Continuous directed evolution has not been demonstrated with this system. Possibly they mean to express that the system supports both ON and OFF selection. Consider rephrasing/expanding to clarify.

Line 140: A closing bracket is missing after TBwtG.

Line 245: The referenced Figure 2A shows no workflow; probably it actually refers to Figure 1B.

Line 276: The second reference for inversion of LacI binding logic (DOI 10.1006/jmbi.1996.0479), describing a large-scale mutational screening, mentions inverted Lac repressors but does not report new genotypes leading to this phenotype. More suitable references might be DOI 10.1016/j.cell.2011.06.035 and 10.1093/nar/gkw125, both of which report directed evolution and characterization of inverted logic Lac repressors.

Line 295: Here it says that no transcriptional activators have been used as components of yeast synthetic transcriptional activators so far, which contradicts definitions given in lines 37/38. Consider replacing with 'prokaryotic activating transcription factor' or similar for clarity.

Lines 422-424: Extended auxotrophy markers were used for increased plasmid integration efficiency. If possible, a reference for this method should be given, or, if unpublished, more information (sequences of extended markers and possibly evidence for intended effect) should be provided.

Reviewer #2:

Remarks to the Author:

General comments:

Eukaryotic genetic switches are far less studied when compared to those of prokaryotes. This manuscript demonstrated direct evolution and engineering of yeast genetic switches using a hybrid selector protein consisting of a negative selector (hsvTK23), a positive selector (Ble), and a fluorescent protein reporter (GFP). Using the hybrid selector, the authors constructed various yeast genetic switches with high-quality and unique functions. The resulting genetic switches were utilized to construct a quorum sensing sensor and an AND-gated carotenoid biosynthesis in yeast. The strategies proposed in this paper would advance yeast synthetic biology and metabolic engineering. Potential impacts and benefits of this kind of publication come from wide usages of the developed platform for various applications. This reviewer strongly suggests to submit the constructed plasmids in this study into a public depository, such as Addgene.

Specific comments:

1. Line 40-42: please provide references for this claim.
2. Line 242: "indicating too much CamTA expression" is redundant and can be removed or replaced.
3. Line 336-337: explain why the leakiness of ptetO7 presented only when it was applied to CrtYB but was not observed when evaluated with GFP (Fig. 5B)?
4. Instead of a summary of the work, more comparisons with related studies should be included in the discussion section to better demonstrate the significance or unresolved issues of this study.

1. General response

We would like to appreciate two reviewers for their valuable comments to our manuscript. A main point of comments by two reviewers was the comparative description of the advantage/disadvantage of proposed selection platform and their deliverables. We have revised the manuscript according to their suggestions (the major changes are listed below).

Because our package became huge now, we thoroughly re-checked the entire manuscript and samples, where we have noticed one minor mistake in supplementary Figure that must be corrected. Re-sequencing of the expression plasmid of the 1-11E+F109L (appeared in the original **Supplementary Figure 13D**) revealed that it did not contain documented mutations in its promoter and terminator sequences. Therefore, we re-constructed the plasmid with the exact sequence documented in the manuscript and subjected to repeated the experiment again (please see below). Please note that the new experimental data does not change any of our discussion in this paper. Please also note that we did not find any other mistakes in the documents. Now, we are very sure that the revised manuscript is free from errors in data, sequence, and sample names.

Original Figure S13D

Corrected Figure S13D

2. List of major changes

- As mentioned above, the 1-11E+F109L expression plasmid has been reconstructed and reevaluated. Along with this change, **Supplementary Figure 13D** was replaced with new data.
- In response to the Reviewer #1 comment, we tested whether the improved Tet-ON variant (Tet-ON_{K8N, L131L}) could be enriched from an approximately 1:2000 background (with wild-type rTetTA or empty vector) (**Supplementary Figure 17**). We added the new figure as a supplementary, as well as the corresponding description to the Discussion section.
- In response to the Reviewer #1 comment, we determined that the frequency of selection escapes from our negative selection (**Supplementary Table 5**). We added the new table as a supplementary, and the related description to the Discussion section.
- In response to the Reviewer #1 comment, we added a new table (**Supplementary Table 6**), in which the library size for each directed evolution is presented.
- In response to the Reviewer #1 comment, we added the sequence of newly created auxotrophic markers (*URA3red*, *HIS3red*, *LEU2red* and *MET15red*) to **Supplementary Table 3**. Plus, because the sequence of the actual integration site is not fully confirmed, we removed the related notes from **Figure 5**, **Supplementary Figures 1, 4**, and **21**, in case they could be misleading.

- Upon receiving the Reviewer #2 comment, which suggests to clarify the reason why the leakiness of p_{tetO7} was observed when it applied to control carotenoid biosynthesis, we evaluated whether leaky GFP expression appeared when the upstream sequence of p_{tetO7} was changed. We added this data (**Supplementary Figure 18**), and the relevant description to the Discussion section.

3. Point by point responses

REVIEWER COMMENTS (Author's responses are shown in blue)

Reviewer #1 (Remarks to the Author):

Synthetic biology's increasingly complex genetic circuitry requires orthogonal genetic switches. The limited repertoire of suitable, well-characterized switches is still a major bottleneck in the field, particularly in eukaryotic systems. Tominaga et al. address this problem by having developed a selection system for evolving switchable transcription factors in yeast. Selecting for switchable function is a considerably more challenging problem than e.g. the selection of a constitutive binder: Selections for both the active (ON) and the inactive (OFF) state have to be performed and balanced in stringency to enrich genetic switches from a background of constitutively active or inactive variants. The presented selection system achieves this dual selection with a trifunctional fusion protein consisting of a positive (Ble) and a negative selector (hsvTK) protein, and a GFP to facilitate easy characterization of selected variants. The usefulness of the method is demonstrated by a variety of different examples of directed evolution for switchable transcription factors; the authors managed to substantially improve the switching behavior of two synthetic transcription factors, inverted the switching logic of one of them, and engineered a LuxR based yeast genetic switch. Evolved switches were used to construct a carotenoid synthesis pathway with AND control. This article is well-written and the presented research of notable interest to the field of eukaryotic synthetic biology, as a comparable dual selection system does not yet exist in yeast. The variety of presented use cases shows the general robustness of the system. For the reader interested in using the selection system, more quantitative characterizations would be useful to guide the design of selection experiments.

- Specifically, I would suggest doing mock selections of switchers from backgrounds of non-switchers to determine enrichment factors with typical selection parameters.

(Our response)

Thank you for your suggestion. We tested whether the improved Tet-ON variant (Tet-ON_{K8N, L131L}) could be enriched from an approximately 1:2000 background (with wild-type rTetTA or empty vector) (**Supplementary Figure 17**). We added the related description to the Discussion section, as shown below.

Unlike the mock selection, where the functional genetic switches outcompete non-functional (always-ON and -OFF) variants, the actual genetic switch library consists of variants with different characteristics. In most cases, libraries are dominated by wild-type-like (weakly inducible) variants, and it is very important to find the appropriate selection condition that can selectively enrich rare mutants with slightly improved performance. To test whether our selection platform can be applied to real-world applications, we selected an improved Tet-ON switch variant (Tet-ON_{K8N, L131L}) against an approximately 1:2000 background (with wild-type rTetTA or empty vector). Surprisingly, despite the limited difference in the performance of these variants, only a single round of OFF/ON selection yielded a 133-fold enrichment of Tet-ON_{K8N, L131L} over the background (**Supplementary Figure 17**). This result highlights the robustness of the platform.

(B)

Relative abundance of each variants [%]

variant	Run-1		Run-2		Run-3	
	Before	After	Before	After	Before	After
rtTA _{K8N, L131L}	0.047	6.25 (1/16)	0.038	N. D. (0/16)	0.053	6.25 (1/16)
rtTA	48.1	81.3 (13/16)	42.7	81.3 (13/16)	44.7	25.0 (4/16)
empty	51.9	12.5 (2/16)	57.2	18.8 (3/16)	55.2	68.8 (10/16)

Enrichment factor for rtTA_{K8N, L131L}

Run-1	Run-2	Run-3
133	N. D.	118

Supplementary Figure 17. Enrichment for the mutant Tet-ON switch (Tet-ON_{K8N, L131L}) from mock switch library. (A) Yeast cells harboring plasmid with wild-type Tet-ON or Tet-ON_{K8N, L131L}, and empty vector was mixed at a ratio of approximately 1:10⁻³:1. The actual abundance of Tet-ON_{K8N, L131L} was evaluated by counting the colony forming units of each yeast culture. The resultant cell culture was subjected to OFF/ON selections with the same condition of Run-11 as described in **Figure 3**. The same selection experiments were independently performed three times. Colonies were isolated from the resultant three cell populations and the Dox-induced GFP expression was evaluated using flow-cytometry (B). The Tet-ON_{K8N, L131L} variants in the selected cell pools were identified as the variants exhibiting Dox-induced GFP expression comparable to that of the control (not mixed) strain. Variants identified as cells harboring plasmid expressing rtTA_{K8N, L131L} or wild-type rtTA, and empty vector are denoted by black, grey, and green arrows, respectively.

- Further, the system uses a negative selection dependent on the function of two genes (*hsvTK* and *hENT1*), i.e. a loss of function of either of these should lead to false positives. The frequency of spontaneous occurrence of escapers should be assessed, and possibly the mode of escape analyzed.

(Our response)

We understand the reviewer's concerns. As he/she correctly suggests, negative selection dependent on two genes could be a small drawback of our system. We thought it is a good idea to experimentally address this question. We determined the frequency of mutants that escaped from our negative selection (**Supplementary Table 5**). We found that most escapees either lost the *hENT1* gene or acquired mutation(s) in the *hENT1* gene. Thank you for this new insight, we integrated the additional *hENT1* expression cassette into another chromosome and found that the frequency of such mutants can be reduced to $< 10^{-5}$. The related description has now been added to the Discussion section as shown below. We thank the reviewer for this insightful suggestion.

A possible drawback of our selection platform is the requirement of two genes for OFF selection. In theory, loss-of-function mutations could occur both in *hsvtk* or *hENT1* and they would lead to the selection escapees. Indeed, we observed both 5FdU^r- and dP^r-resistant clones with a frequency of 10^{-3} (**Supplementary Table 5**), which were higher than those reported for other systems^{71, 72}. To reduce this mutation-based selection escape, duplication of negative selector genes is effective⁷³. The chromosomal addition of another *hENT1* expression cassette resulted in a >100-fold reduction of escapees (**Supplementary Table 5**). This indicates that the major source of selection escapees emerges via mutations at *hENT1*, not *hsvtk*. It is likely that the *hENT1* expression is toxic to yeast, thereby imposing selective pressure on its loss.

Supplementary Table 5. The frequency of OFF-selection escapees.

Strain + plasmid	N	Frequency of 5FdU ^r colony [-]	Frequency of dP ^r colony [-]
BY4741- hENT1 - p_{tetO7} - TBG + pGK415-rTetTA	1	4×10^{-4}	4×10^{-4}
	2	2×10^{-3}	1×10^{-3}
	3	8×10^{-4}	5×10^{-4}
	Ave \pm SD	$1 \times 10^{-3} \pm 7 \times 10^{-4}$	$8 \times 10^{-4} \pm 5 \times 10^{-4}$
BY4741- hENT1 - p_{tetO7} - TBG/hE NT1 + pGK415-rTetTA	1	8×10^{-6}	6×10^{-6}
	2	2×10^{-6}	2×10^{-6}
	3	$< 4 \times 10^{-7}$	$< 4 \times 10^{-7}$

5FdU^r: 5FdU resistant, dP^r: dP resistant

- In addition, it might be helpful to have a supplementary table of the actual library sizes of individual selection experiments since they varied by two orders of magnitude. This would allow conclusions about the expected frequency of desired phenotypes in random libraries and could provide researchers attempting similar selections valuable guidance as to how large libraries should be. Notably, for several selections small library sizes (10^3) apparently were sufficient.

(Our response)

Thank you for your comment. We have added a new table (**Supplementary Table 6**), where the library size for each directed evolution is presented.

Supplementary Table 6. Library size for each directed evolution experiment.

Genetic Switch	Randomized DNA	Estimated library size (Number of individual clones)	Related Figure
Tet-ON	rTetTA cassette	2×10^5	Figure 3
DAPG-OFF	PhITA cassette	1×10^4	Figure 4B, S9A
Camphor-OFF	CamTA cassette	2×10^3	Figure 4C, S10
DAPG-ON (1 st round)	PhITA cassette	1×10^4	Figure 4D, S13A
DAPG-ON (2 nd round)	rPhITA cassette	Not determined	Figure 4D, S13C
HSL-ON (1 st round)	LuxTA cassette	2×10^4	Figure 4F, S15A
HSL-ON (2 nd round)	LuxTA cassette	Not determined	Figure 4F, S15B

- In the discussion regarding FACS-based selection it is said that “the appropriate selection threshold for a given experiment is virtually impossible to guess a priori. Lacking the ability to predict what type of mutants will emerge or at what probabilities, experimentalists are required to repeat the sorting experiments until the right gating condition happen to be achieved.” This reasoning of the benefits of the presented selection system over FACS-based selections does not seem compelling, because i) appropriate controls with known activation strength (e.g. low, medium, strong) can be used to inform the FACS experimentalist of appropriate gating conditions and ii) the presented system faces the same issue, which the authors remedy by running parallel selections with different selection stringencies. This is perfectly doable with FACS-based selection as well. There are other downsides of FACS-based selections which could make for more convincing arguments, e.g. high signal scatter for low fluorescence signals causing difficulties during “OFF” selection.

(Our response)

Thank you for your comment. As the reviewer pointed out, sorting experiments with different gating conditions can be performed with FACS; however, their throughput is largely limited compared with our system. Unlike the FACS-based selection, a thousand selection experiments can be performed in a multi-well format (it is not realistic to operate multiple FACS-devices). In addition, by increasing the number of sorted cells to minimize the signal variation within clones, the throughput of FACS will further decrease. This could be a drawback of FACS, as the reviewer pointed out. In summary, one of the main claims of this paper lies in the importance of parallel testing of the selection conditions with unprecedented throughput (with liquid handling devices, one can conduct tens of thousands of selections using 384- or 96-well plates). To clarify this point, we have revised the manuscript as follows:

Classical agar **plate**-based ON and OFF selection systems, such as His3p/3-aminotriazole (3-AT) and Ura3p/5-fluoroorotic acid (5-FOA), fail to discriminate **between** improved and prototypic genetic switches when their output levels under selection are below or above the given selection threshold^{19, 20}. This challenge requires researchers to reconstruct the expression levels of selectors in individual contexts and/or at each round of evolutionary cycling. Even with the FACS-based ON and OFF selections where the selection threshold can be easily modulated by changing the sorting threshold^{17, 18}, experimentalists are required to repeat the sorting experiments until the **correct** gating condition is achieved. **Instead of repeating, for instance, the multiple FACS sorting experiments, our selection platform allows to operate the multiple selections in a multi-well format, with different selection conditions. Given that the processes described here can be implemented by liquid-based operation, our workflow can be fully automated and can be combined with recently developed *in vivo* mutagenesis technologies^{68, 69} to further accelerate the evolutionary cycle of yeast genetic switches.**

- Further, I am somewhat skeptical of the claim that the system “can be combined with recently developed *in vivo* mutagenesis technologies to conduct continuous evolution”. If mutagenesis is ongoing during both ON and OFF selections it is to be expected that this does not facilitate enrichment of switchers, but spontaneous emergence of variants with high fitness in the respective environment during each selection cycle. Alternatively, mutagenesis could be performed only before dual selection, but this effectively equates to ‘conventional’ library generation and is not ‘continuous evolution’ anymore. However, if the authors have a promising strategy to employ their system in a continuous evolution approach with *in vivo* mutagenesis, they should briefly outline it here.

(Our response)

Thank you for your comment. 'Conduct continuous evolution' was changed to 'further accelerate the evolutionary cycle of yeast genetic switches' in the revised manuscript.

I have further specific comments on the manuscript:

- Line 65: It is somewhat unclear what the authors mean by ‘continuous selection’. Continuous directed evolution has not been demonstrated with this system. Possibly they mean to express that the system supports both ON and OFF selection. Consider rephrasing/expanding to clarify.

(Our response)

Thank you for your comment. It has been corrected to 'seamless OFF/ON selection' in the revised manuscript.

- Line 140: A closing bracket is missing after TBwtG.

(Our response)

We thank the reviewer for pointing this out. This has been corrected in the revised manuscript.

- Line 245: The referenced Figure 2A shows no workflow; probably it actually refers to Figure 1B.

(Our response)

We thank the reviewer for pointing this out. This has been corrected to Figure 1B in the revised manuscript.

- Line 276: The second reference for inversion of LacI binding logic (DOI 10.1006/jmbi.1996.0479), describing a large-scale mutational screening, mentions inverted Lac repressors but does not report new genotypes leading to this phenotype. More suitable references might be DOI 10.1016/j.cell.2011.06.035 and 10.1093/nar/gkw125, both of which report directed evolution and characterization of inverted logic Lac repressors.

(Our response)

We thank the reviewer for pointing this out. This has been corrected in the revised manuscript.

- Line 295: Here it says that no transcriptional activators have been used as components of yeast synthetic transcriptional activators so far, which contradicts definitions given in lines 37/38. Consider replacing with 'prokaryotic activating transcription factor' or similar for clarity.

Thank you for your comment. This has been corrected to 'prokaryotic transcriptional activators' in the revised manuscript.

- Lines 422-424: Extended auxotrophy markers were used for increased plasmid integration efficiency. If possible, a reference for this method should be given, or, if unpublished, more information (sequences of extended markers and possibly evidence for intended effect) should be provided.

(Our response)

Thanks for the comment. The sequence of these markers is added to **Supplementary Table 3**. Plus, because the sequence of the actual integration site is not fully confirmed, we removed the related notes from **Figure 5**, **Supplementary Figures 1, 4, and 21**, in case they could be misleading.

Reviewer #2 (Remarks to the Author):

General comments:

Eukaryotic genetic switches are far less studied when compared to those of prokaryotes. This manuscript demonstrated direct evolution and engineering of yeast genetic switches using a hybrid selector protein consisting of a negative selector (hsvTK²³), a positive selector (Ble), and a fluorescent reporter (GFP). Using the hybrid selector, the authors constructed various yeast genetic switches with high-quality and unique functions. The resulting genetic switches were utilized to construct a quorum sensing sensor and an AND-gated carotenoid biosynthesis in yeast. The strategies proposed in this paper would advance yeast synthetic biology and metabolic engineering. Potential impacts and benefits of this kinds of publication come from wide usages of the developed platform for various applications. This reviewer strongly suggests to submit the constructed plasmids in this study into a public depository, such as Addgene.

(Our response)

We thank the reviewer for the suggestion of depositing our plasmids into the public repository. We have completed an online submission of our plasmids into Addgene and they will be ready for distribution as soon as they pass the quality-control process.

Specific comments:

1. Line 40-42: please provide references for this claim.

(Our response)

Thank you for your comment. We have added the following three references and the relevant modifications in the sentence.

1. Roney et al. (2016) described that excess expression of synthetic transcription factor resulted in the reduced fold-induction.
2. Loew et al., (2010) described that the length of interval sequence between operators affect the inducibility.
3. Ede et al., (2016) [DOI: doi: 10.1021/acssynbio.5b00266] reported that gene induction properties could vary depending on the core promoter.

2. Line 242: "indicating too much CamTA expression" is redundant and can be removed or replaced.

(Our response)

Thank you for your comment. This has been removed from the revised manuscript.

3. Line 336-337: explain why the leakiness of ptetO7 presented only when it was applied to CrtYB but was not observed when evaluated with GFP (Fig. 5B)?

(Our response)

Thank you for your comment. We related this to the context-dependency of p_{tetO7} . To clarify this point, we evaluated whether leaky GFP expression from p_{tetO7} was observed when the upstream sequence (reverse-oriented *TDH3* promoter in this case) used for carotenoid biosynthesis was used. At this time, we found significant leakiness. We added this data (**Supplementary Figure 18**) and the relevant discussion to the Discussion section, as shown below.

The yeast promoters are known to be affected by their surrounding sequences⁷⁴, and the level of leakiness of synPs appears to be quite different depending on the reporter type (phenotype). Indeed, we observed the leakiness of the Tet-ON switch when it was applied to control the carotenoid biosynthesis gene *crtYB* (**Fig. 5D**), while no detectable leakiness was observed for GFP expression (**Fig. 5B**). This could be simply because even the least expression of CrtYB results in pigment biosynthesis, or it could be from a transcription read-through from the upstream sequence. Evidently, adding this sequence to *p_{tetO7}* caused significant leakage of the GFP expression (**Supplementary Figure 18**). A set of terminators and ribozyme sequences could be used to insulate synPs from their upstream sequences to block their transcription read-through⁷⁵.

4. Instead of a summary of the work, more comparisons with related studies should be included in the discussion section to better demonstrate the significance or unresolved issues of this study.

(Our response)

Thank you very much for your comments. According to the suggestion, the description in the Discussion section was entirely revised to include the comparison of our selection platform and their deliverable genetic switches with the existing systems.

- First, we describe the advantage of our selection platform over conventional FACS-based selection. Our selection is featured with its unprecedented throughput, enabling us to perform a thousand selection experiments in a multi-well format (**first paragraph**).

- Second, we added a description of the critical drawbacks of our selection platform and its solution. As the other reviewer pointed out, negative selection dependent on two genes could be a small drawback of our system. We showed that doubling the hENT1 expression cassette could be a solution for this (**fourth paragraph**).

- Finally, we added the comparison of the genetic switches developed in this study against the existing yeast genetic switches. Most of our switches showed over 100-fold GFP induction upon the addition of the corresponding inducers. This performance is comparable to that of recently reported extensively optimized yeast genetic switches (Y. Chen *et al.*, *Nat. Microbiol.*, **5**, pages1349–1360(2020) (<https://doi.org/10.1038/s41564-020-0757-2>); **sixth paragraph**).

3. Discussion

Classical agar plate-based ON and OFF selection systems, such as His3p/3-aminotriazole (3-AT) and Ura3p/5-fluoroorotic acid (5-FOA), fail to discriminate between improved and prototypic genetic switches when their output levels under selection are below or above the given selection threshold^{19, 20}. This challenge requires researchers to reconstruct the expression levels of selectors in individual contexts and/or at each round of evolutionary cycling. Even with the FACS-based ON and OFF selections where the selection threshold can be easily modulated by changing the sorting threshold^{17, 18}, experimentalists are required to repeat the sorting experiments until the correct gating condition is achieved. Instead of repeating, for instance, the multiple FACS sorting experiments, our selection platform allows to operate the multiple selections in a multi-well format, with different selection conditions. Given that the processes described here can be implemented by liquid-based operation, our workflow can be fully automated and can be combined with recently developed *in vivo* mutagenesis technologies^{68, 69} to further accelerate the evolutionary cycle of yeast genetic switches.

Using the workflow described in **Figs. 1 and 3A**, we isolated multiple variants with inverted switching behaviours (DAPG-ONs, **Fig. 4D**), novel HSL sensors (HSL-ONs, **Fig. 4F**), and variants of the HSL sensors with improved HSL sensitivities (**Fig. 4F**). The genetic switches described here might need to be re-optimized in other contexts because of the intrinsic context-dependency⁷⁰, it should be possible to quickly re-adapt them to new contexts as was described in this paper. Indeed, one of the best-engineered sTAs, TetTA, first malfunctioned in our system, exhibiting low stringency (**Fig. 2**). Similarly, two other previously reported sTAs, PhlTA and CamTA, were virtually non-functional in the context tested in the present study, and one of these published switches (PhlTA) exhibited unexpected and severe cell toxicity (**Supplementary Fig. 9**). Nonetheless, a single round of randomization and OFF/ON selections rapidly recovered the reported behaviour in the new context.

Unlike the mock selection, where the functional genetic switches outcompete non-functional (always-ON and -OFF) variants, the actual genetic switch library consists of variants with different characteristics. In most cases, libraries are dominated by wild-type-like (weakly inducible) variants, and it is very important to find the appropriate selection condition that can selectively enrich rare mutants with slightly improved performance. To test whether our selection platform can be applied to real-world applications, we selected an improved Tet-ON switch variant (Tet-ON_{K8N, L131L}) against an approximately 1:2000 background (with wild-type rTetTA or empty vector). Surprisingly, despite the limited difference in the performance of these variants, only a single round of OFF/ON selection yielded a 133-fold enrichment of Tet-ON_{K8N, L131L} over the background (**Supplementary Figure 17**). This result highlights the robustness of the platform.

A possible drawback of our selection platform is the requirement of two genes for OFF selection. In theory, loss-of-function mutations could occur both in *hsvtk* or *hENT1* and they would lead to the selection escapees. Indeed, we observed both 5FdU- and dP-resistant clones with a frequency of 10^{-3} (**Supplementary Table 5**), which were higher than those reported for other systems^{71, 72}. To reduce this mutation-based selection escape, duplication of negative selector genes is effective⁷³. The chromosomal addition of another *hENT1* expression cassette resulted in a >100-fold reduction of escapees (**Supplementary Table 5**). This indicates that the major source of selection escapees emerges via mutations at *hENT1*, not *hsvtk*. It is likely that the hENT1 expression is toxic to yeast, thereby imposing selective pressure on its loss.

The yeast promoters are known to be affected by their surrounding sequences⁷⁴, and the level of leakiness of synPs appears to be quite different depending on the reporter type (phenotype). Indeed, we observed the leakiness of the Tet-ON switch when it was applied to control the carotenoid biosynthesis gene *crtYB* (**Fig. 5D**), while no detectable leakiness was observed for GFP expression (**Fig. 5B**). This could be simply because even the least expression of CrtYB results in pigment biosynthesis, or it could be from a transcription read-through from the upstream sequence. Evidently, adding this sequence to *p_{tetO7}* caused significant leakage of the GFP expression (**Supplementary Figure 18**). A set of terminators and ribozyme sequences could be used to insulate synPs from their upstream sequences to block their transcription read-through⁷⁵.

Given the shortage of reliable tools, yeast synthetic biology has provided limited capacity for metabolic control. Using our platform, we created dozens of inducible genetic switches with distinct specifications. We tested whether these novel switches could be independently used to control pathway

genes in one cell (**Fig. 5D**). Among the novel switches, **the performance of** the DAPG-ON switch was comparable to **that of** the Tet-ON switch, which is known as one of the best switches in terms of its fold induction ($>10^2$) (**Fig. 5B**), **as well as that of the recently reported high-performing switches**⁷⁵. We then demonstrated that these evolved switches could be further improved in fold induction simply by tweaking the numbers of operators¹⁰ tandemly aligned in synPs. Specifically, increasing the number of *luxO* repeats from 5 to 10 (p_{luxO10} instead of p_{luxO5}) greatly improved the fold change of the HSL-ON switch to levels comparable to those obtained with both DAPG-ON and Tet-ON switches (**Supplementary Fig. 19**). In summary, our workflow enables **the** *in situ* construction of high-quality genetic switches **that are** useful in various synthetic biology applications.

Reviewers' Comments:

Reviewer #1:

Remarks to the Author:

The authors have adequately addressed all major points risen in the first round of review, and I feel the manuscript has been substantially improved.

I have only a few minor comments/suggestions:

- In the figure caption of Supplementary Figure 17, I suggest changes for clarity of attribution, e.g.:

The Tet-ONK8N, L131 variants in the selected cell pools were identified by a Dox-inducible GFP expression pattern comparable to that of the Tet-ONK8N, L131 control.

Variants identified as cells harboring plasmid expressing rtTAK8N, L131L or wild-type rtTA, and empty vector are denoted by green, gray, and black arrows, respectively. (instead of: "are denoted by black, grey, and green arrows")

- Also in Supplementary Figure 17B, I suggest having the controls in the graph in the same order as in the table (rtTAK8N, L131L; wild-type rtTA; empty vector), and possibly labelled in the graph. Further, the table caption should read "Relative abundance of each variant [%]"

- in line 38 "eukaryotes" is capitalised

- in lines 390, 394 and 401 there is a space before reference numbers

Reviewer #2:

Remarks to the Author:

The authors made revisions appropriately according to reviewers' suggestions.

General Response

We thank all the reviewers for their thoughtful comments. We have addressed their suggestions as described below. We have also revised all related documents to comply with the journal policies. All the corresponding changes in main text file are highlighted in red in the revised manuscript. Any track changes in supplementary information files are removed to comply with the journal policies that they will be uploaded with the published article as they are submitted with the final version of our manuscript.

REVIEWERS' COMMENTS

Reviewer #1 (Remarks to the Author):

The authors have adequately addressed all major points risen in the first round of review, and I feel the manuscript has been substantially improved.

We appreciate your willingness to review and help to improve our manuscript.

I have only a few minor comments/suggestions:

- In the figure caption of Supplementary Figure 17, I suggest changes for clarity of attribution, e.g.: The Tet-ONK8N, L131 variants in the selected cell pools were identified by a Dox-inducible GFP expression pattern comparable to that of the Tet-ONK8N, L131 control. Variants identified as cells harboring plasmid expressing rtTAK8N, L131L or wild-type rtTA, and empty vector are denoted by green, gray, and black arrows, respectively. (instead of: "are denoted by black, grey, and green arrows")
- Also in Supplementary Figure 17B, I suggest having the controls in the graph in the same order as in the table (rtTAK8N, L131L; wild-type rtTA; empty vector), and possibly labelled in the graph. Further, the table caption should read "Relative abundance of each variant [%]"

Response

Thank you for your helpful suggestions. We revised the Supplementary Figure 17 and its legend accordingly as shown below.

Supplementary Figure 17. Enrichment for the mutant Tet-ON switch (Tet-ON_{K8N, L131L}) from mock switch library. (a) Yeast cells harboring plasmid with wild-type Tet-ON or Tet-ON_{K8N, L131L}, and empty vector was mixed at a ratio of approximately 1:10⁻³:1. The actual abundance of Tet-ON_{K8N, L131L} was evaluated by counting the colony forming units of each yeast culture. The resultant cell culture was subjected to OFF/ON selections with the same condition of Run-11 as described in **Figure 3**. The same selection experiments were independently performed three times. Colonies were isolated from the resultant three cell populations and the Dox-induced GFP expression was evaluated using flow-cytometry (b). The Tet-ON_{K8N, L131L} variants in the selected cell pools were identified by a Dox-inducible GFP expression pattern comparable to that of the Tet-ON_{K8N, L131L} control. The bars represent the ratio of the signal of the ON/OFF state (only for controls, N=3. Error bars represent the SD of three independent experiments). Variants identified as cells harboring plasmid expressing *rtetTA*_{K8N, L131L} or wild-type *rtetTA*, and empty vector are denoted by green, grey, and black arrows, respectively.

- in line 38 “eukaryotes” is capitalised
- in lines 390, 394 and 401 there is a space before reference numbers

Response

Thank you for your helpful comment. We corrected these points in the revised manuscript.

Reviewer #2 (Remarks to the Author):

The authors made revisions appropriately according to reviewers' suggestions.

Response

We appreciate the reviewer's help to improve our manuscript.